# Multi-agent Dynamic Algorithm Configuration

**Ke Xue**[1]*, **Jiacheng Xu**[1]*, **Lei Yuan**[1,2],
**Miqing Li**[3], **Chao Qian**[1]†, **Zongzhang Zhang**[1], **Yang Yu**[1,2]
[1] State Key Laboratory for Novel Software Technology, Nanjing University
[2] Polixir Technologies
[3] CERCIA, School of Computer Science, University of Birmingham
{xuek, xujc, yuanl}@lamda.nju.edu.cn,
m.li.8@bham.ac.uk, {qianc, zzzhang, yuy}@nju.edu.cn

## Abstract

Automated algorithm configuration relieves users from tedious, trial-and-error tuning tasks. A popular algorithm configuration tuning paradigm is dynamic algorithm configuration (DAC), in which an agent learns dynamic configuration policies across instances by reinforcement learning (RL). However, in many complex algorithms, there may exist different types of configuration hyperparameters, and such heterogeneity may bring difficulties for classic DAC which uses a single-agent RL policy. In this paper, we aim to address this issue and propose multi-agent DAC (MA-DAC), with one agent working for one type of configuration hyperparameter. MA-DAC formulates the dynamic configuration of a complex algorithm with multiple types of hyperparameters as a contextual multi-agent Markov decision process and solves it by a cooperative multi-agent RL (MARL) algorithm. To instantiate, we apply MA-DAC to a well-known optimization algorithm for multi-objective optimization problems. Experimental results show the effectiveness of MA-DAC in not only achieving superior performance compared with other configuration tuning approaches based on heuristic rules, multi-armed bandits, and single-agent RL, but also being capable of generalizing to different problem classes. Furthermore, we release the environments in this paper as a benchmark for testing MARL algorithms, with the hope of facilitating the application of MARL.

## 1 Introduction

Finding right configurations of hyperparameters is critical for many learning and optimization algorithms [18]. Automated methods, such as algorithm configuration (AC) [17, 27], emerge to search for right configurations, with the aim of relieving users from tedious, trial-and-error tuning tasks. However, static configuration policies obtained by AC may not necessarily yield optimal performance (compared with dynamic policies) since algorithms may require different configurations at different stages of their execution [38].

Dynamic algorithm configuration (DAC) [3, 1] is a well-known paradigm for obtaining dynamic configuration policies. Unlike AC, DAC can dynamically adjust the algorithm's configuration during the optimization process, through formulating it as a contextual Markov decision process (MDP) and then solving it by reinforcement learning (RL) [33]. DAC has been found to outperform static methods on many tasks, including the learning rate tuning of deep neural networks [7], step-size adaptation of evolution strategies [35], and heuristic selection of AI planning [38].

---

*Equal Contribution
†Corresponding Author

The task of DAC typically focuses on a single type of configuration hyperparameter, such as tuning the step-size in CMA-ES [35]. However, due to the increasing complexity of real-world problem modeling, there are many algorithms whose performance rests on multiple types of hyperparameters. Tuning one type while fixing the rest may not produce promising results. Take a popular evolutionary algorithm for multi-objective optimization problems, MOEA/D [51], as an example. It has four types of configuration hyperparameters: weights, neighborhood size, reproduction operator type, and parameters associated with the reproduction operator used. All of them are critical and significantly affect the performance of the algorithm [44]. Finding a near-optimal configuration combination for each part of these hyperparameters requires massive manual effort [12]. However, how to jointly adjust multiple types of configuration hyperparameters is still an open problem [1].

In this paper, we attempt to extend DAC to deal with tasks with multiple types of configuration hyperparameters. We model this as a cooperative multi-agent problem [29], with each agent handling one type of hyperparameter, for a shared goal (i.e., maximizing the team reward). Specifically, the proposed multi-agent DAC (MA-DAC) method formulates the task as a contextual multi-agent MDP (MMDP) [5], and solves it by a common pay-off cooperative multi-agent RL (MARL) algorithm. As an instantiation, we apply MA-DAC to the multi-objective evolutionary algorithm MOEA/D [51] to learn the right configurations of its four types of hyperparameters by the classic MARL algorithm called value-decomposition networks (VDN) [40].

Experiments on well-established multi-objective optimization problems show that MA-DAC outperforms other configuration tuning methods based on heuristic rules [31], multi-armed bandits [21, 15] and single-agent RL [28]. Furthermore, we demonstrate the generalization ability of the learned MA-DAC policy to both inner classes (different instances with the same number of objectives) and outer classes (different instances with different numbers of objectives). Ablation studies also demonstrate the importance of tuning every type of hyperparameter.

Our contributions are three-fold:

1. To the best of our knowledge, the MA-DAC method is the first one to address the dynamic configuration of algorithms with multiple types of hyperparameters.

2. The contextual MMDP formulation of MA-DAC is analyzed, and experimental results show that the presented formulation works well and has good generalization ability.

3. The instantiation of configuring MOEA/D in this work can be used as a benchmark problem for MARL. The heterogeneity of MOEA/D's hyperparameters and the stochasticity of its search can promote the research of MARL algorithms. Besides, the learned policies are useful for a specific type of optimization task - multi-objective optimization, which will facilitate the application of MARL.

## 2 Background

### 2.1 Dynamic algorithm configuration

Different from the static configuration of AC, DAC aims at dynamically adjusting the configuration hyperparameters of an algorithm during its optimization process. Biedenkapp et al. [3] formulated DAC as a contextual MDP $\mathcal{M}_{\mathcal{I}} := \mathcal{M}_{i \sim \mathcal{I}}$ and applied RL to solve it. $\mathcal{I}$ represents the space of problem instances, and each $\mathcal{M}_i := \langle \mathcal{S}, \mathcal{A}, \mathcal{T}_i, r_i \rangle$ [3, 10] corresponds to one target problem instance $i \in \mathcal{I}$. The notion of context $\mathcal{I}$ allows to study generalization of policies in a principled manner [20]. Given a target algorithm $A$ with its configuration hyperparameters space $\Theta$, a DAC policy $\pi \in \Pi$ maps the state $s \in \mathcal{S}$ (e.g., history of changes in the objective value achieved by the target algorithm $A$) to the action $a \in \mathcal{A}$ (i.e., a hyperparameter $\theta \in \Theta$ of the target algorithm $A$). DAC aims at improving the performance of $A$ on a set of instances (e.g., optimization functions). Given a probability distribution $p$ over the space $\mathcal{I}$ of problem instances, the objective of DAC is to find an optimal policy $\pi^*$. That is,

$$\pi^* \in \arg\min_{\pi \in \Pi} \int_{i \in \mathcal{I}} p(i)c(\pi, i)\mathrm{d}i, \tag{1}$$

where $i \in \mathcal{I}$ is an instance to be optimized, and $c(\pi, i) \in \mathbb{R}$ is the cost function of the target algorithm with policy $\pi$ on the instance $i$. It has been shown that DAC outperforms static policies in learning rate

adaptation in SGD [7], step-size adaptation in CMA-ES [35], and heuristic selection in planning [38]. These applications all only involve a single type of hyperparameter. Dynamic configurations of complex algorithms with multiple types of hyperparameters have been found to be difficult for current DAC methods [10, 1].

## 2.2  Multi-agent reinforcement learning

A multi-agent system [48] under fully observable cooperative situation can be modeled as an MMDP [5], which can be formalized as $\mathcal{M} := \langle \mathcal{N}, \mathcal{S}, \{\mathcal{A}_j\}_{j=1}^n, \mathcal{T}, r \rangle$, where $\mathcal{N}$ is a set of $n$ agents, $\mathcal{S}$ is the state space, and $\mathcal{A}_j$ is agent $j$'s action space. At each time-step, agent $j \in \mathcal{N}$ acquires $s \in \mathcal{S}$ and then chooses the action $a^{(j)} \in \mathcal{A}_j$. The joint action $\boldsymbol{a} = \langle a^{(1)}, \dots, a^{(n)} \rangle$ leads to next state $s' \sim \mathcal{T}(\cdot \mid s, \boldsymbol{a})$ and all agents get a shared global reward $r(s, \boldsymbol{a})$.

The goal of an MMDP is to find a joint policy that maps the states to probability distributions over joint actions, $\boldsymbol{\pi} : S \rightarrow \Delta(\mathcal{A}_1 \times \mathcal{A}_2 \times \cdots \times \mathcal{A}_n)$, where $\Delta(\mathcal{A}_1 \times \mathcal{A}_2 \times \cdots \times \mathcal{A}_n)$ stands for the distribution over joint actions, with the goal of maximizing the global value function:

$$Q^{\boldsymbol{\pi}}(s, \boldsymbol{a}) = \mathbb{E}_{\boldsymbol{\pi}} \left[ \sum_{t=0}^{\infty} \gamma^t r(s_t, \boldsymbol{a}_t) \mid s_0 = s, \boldsymbol{a_0} = \boldsymbol{a} \right]. \tag{2}$$

## 3  Multi-agent DAC

This section is devoted to the MA-DAC method, where Section 3.1 is concerned with formulating MA-DAC as a contextual MMDP and Section 3.2 explains components of MA-DAC.

### 3.1  Problem formulation

We propose MA-DAC as a new paradigm for solving the dynamic configuration of algorithms with multiple types of hyperparameters. We formulate MA-DAC as a contextual MMDP $\mathcal{M}_{\mathcal{I}} := \{\mathcal{M}_i\}_{i \sim \mathcal{I}}$, where $\mathcal{M}_i := \langle \mathcal{N}, \mathcal{S}, \{\mathcal{A}_j\}_{j=1}^n, \mathcal{T}_i, r_i \rangle$ is a single MMDP as defined in Section 2.2. The notion of context $\mathcal{I}$ induces multiple MMDPs [3, 10]. Each $\mathcal{M}_i$ stands for a specific instance $i$ sampled from a given distribution over $\mathcal{I}$, where an instance $i \in \mathcal{I}$ corresponds to a function $f$ to be optimized. Different $\mathcal{M}_i$s have shared state and action spaces, but with different transitions and reward functions. Algorithms are often tasked with solving different problem instances from the same or similar domains. Searching for well-performing parameter settings on a specific instance might achieve strong performance on that individual instance but might perform poorly in new instances. Therefore, we explicitly take the instance distribution $\mathcal{I}$ as context into account to facilitate generalization [3].

Given a parameterized algorithm $A$ with a configuration space $\Theta = \{\theta_j\}_{j \in \mathcal{N}}$, one agent in MA-DAC is to handle one type of configuration hyperparameter $\theta_j \in \Theta$ and all the agents work with the same goal, i.e., maximizing the team reward. The action space of each agent $j \in \mathcal{N}$ is the hyperparameter space of the corresponding $\theta_j$. Note that MA-DAC can be seen as a common-payoff cooperative MARL problem and can be solved by any cooperative MARL algorithm.

### 3.2  Components of MA-DAC

Next, we introduce several important components of MA-DAC, including the state, action, transition, reward, and instance set.

**State.**   The state is used to describe the situation of the target algorithm, which is a key component in MA-DAC. We suggest that the state should have the following properties.

1. Accessibility. The state should be accessible in each step during the optimization process.
2. Representation. The state should reflect the information in the optimization process.
3. Generalization. The learned policy is expected to generalize to inner and even outer classes of instances. Thus, the state should consist of the common features across different instances.

Besides, it is better if the state can be easily obtained, reducing the computational cost. Note that some of the above properties of state formulation have also been suggested for DAC [24, 10].

**Action.** Each agent in MA-DAC focuses on one type of configuration hyperparameter. Its action is the value of the hyperparameter that should be adjusted, which can be discrete or continuous. MA-DAC agents are heterogeneous because they have completely different action spaces and affect different types of hyperparameters of the target algorithm.

**Transition.** The transition function describes the dynamics of the problem. For an iterative algorithm, each iteration can be defined as a step. Given state $s_t$, each agent $j$ acts $a_t^{(j)}$, and the probability of reaching state $s_{t+1}$ can be expressed as $\mathcal{T}(s_{t+1} \mid s_t, \boldsymbol{a}_t)$. Different from the state and action spaces, the transition function depends on the given instance $f \in \mathcal{F}$. Thus, the policy should identify the characteristics of the current instance and take the optimal action under it.

**Reward.** The reward is used to guide the policy's learning process, whose quality has a significant impact on the policy's final performance. Many RL benchmark environments include a well-defined reward function. However, in many application scenarios, we must define the reward function manually based on a specific metric that can indicate the final performance.

Since the goal of an iterative optimization algorithm at iteration $t$ is to find a solution with the best objective function value so far rather than to just find a better solution than the solution at the $t-1$ step (since the quality of solutions in the last step may be poor), it would make sense to give the policy a reward when finding a better solution than the best solution so far (rather than finding a better solution than the solution at the last step). Another factor one may need to consider when designing a reward function is that with time, it is getting harder to find better-quality solutions. In the beginning, it may be easy to achieve rapid improvement of solution quality, but in later stages, improvement could be harder. As such, a reward function that rewards more an agent who can find a better solution in later stages can encourage the agent to find a very good solution in the end. Considering these two factors, we propose the following reward function at step $t$:

$$r_t = \begin{cases} (1/2) \cdot (p_{t+1}^2 - p_t^2) & \text{if } f(s_{t+1}) < f_t^* \\ 0 & \text{otherwise} \end{cases}, \tag{3}$$

where

$$p_{t+1} = \begin{cases} \frac{f(s_0) - f(s_{t+1})}{f(s_0)} & \text{if } f(s_{t+1}) < f_t^* \\ p_t & \text{otherwise} \end{cases}, \tag{4}$$

and $f_t^*$ is the minimum metric value found until step $t$.

As shown in Figure 1, the isosceles right-angled triangle indicates the maximum return we can get, where $p^*$ is the largest value of $p_t$. At step $t$, the current reward we obtain is the area of the triangle with side length $p_{t+1}$ minus the area of the triangle with side length $p_t$. The proposed triangle-based reward function has two properties: 1) it only rewards the agent when a new solution is better than the best solution found so far and 2) the reward increases in later stages. The comparative experiment is provided in Appendix B.2. This design may be applicable to a wide range of similar tasks.

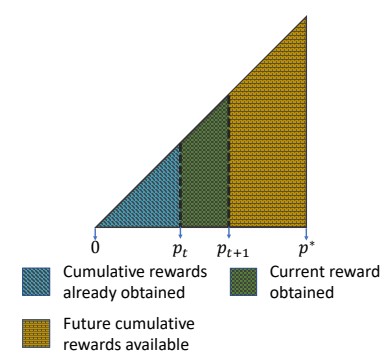

Figure 1: Illustration of the reward function.

**Instance set.** Instance set defines the optimization problem instances that have to be solved by the target algorithm. Note that it is possible to learn policies even across very heterogeneous problem instances by using the context information [3, 35]. Admittedly, if the instances have some similar properties, the learned policy can generalize better [10].

## 4 Applying MA-DAC to dynamic configuration tuning of MOEA/D

As an instantiation, we apply MA-DAC to a well-established evolutionary algorithm for multi-objective optimization problems, multi-objective evolutionary algorithm based on decomposition (MOEA/D) [51]. As one of the most widely used multi-objective evolutionary algorithms, MOEA/D

has several heterogeneous types of configuration hyperparameters (e.g., weights, neighborhood size, and reproduction operator type), which is a good fit for the proposed MA-DAC. We first briefly introduce multi-objective optimization and MOEA/D in Section 4.1, followed by detailed explanations of applying MARL to MOEA/D in Section 4.2, which results in an MARL benchmark, denoted by MaMo. Finally, we compare MaMo with other MARL benchmarks in Section 4.3.

## 4.1 Multi-objective optimization and MOEA/D

Multi-objective optimization refers to an optimization problem with more than one objective to be considered simultaneously. It is very common in the real world. For example, in deep neural networks, apart from the accuracy, one may also care about the model size, latency, and/or energy consumption [26, 14].

A prominent feature of multi-objective optimization problems (MOPs) is that unlike single-objective optimization problems, in MOPs there is no single optimal solution, but a set of Pareto optimal solutions. A solution is called Pareto optimal if there is no other solution in the search space that dominates it[1]. Unfortunately, the number of Pareto optimal solutions for a given MOP is typically prohibitively large or even infinite. Therefore, the goal of a multi-objective optimization algorithm is to find a good approximation that well represents all the Pareto optimal solutions.

Evolutionary algorithms have demonstrated their effectiveness to solve MOPs. Their population-based nature can approximate the problem's optimal solutions within one execution, with each solution in the population representing a unique trade-off among the objectives. MOEA/D [51] is a representative multi-objective evolutionary algorithm. Unlike other mainstream multi-objective evolutionary algorithms like NSGA-II [8], which compare solutions based on their Pareto dominance relation and density in the population, MOEA/D converts an MOP into a number of single-objective sub-problems through a number of weights, where neighboring solutions work cooperatively for the optimal solutions for the single-objective sub-problems. As such, MOEA/D entails different heterogeneous types of hyperparameters, e.g., weights and neighborhood size. Such hyperparameters significantly affect the performance of the algorithm. For example, different weight distributions are suitable for MOPs with different Pareto front shapes as the weights are used to control the distribution of the final population [23]. A large neighborhood has high search ability in the objective space while a small neighborhood is beneficial for diversity maintenance in the decision space [19]. In short, finding right configurations for these hyperparameters requires massive manual effort and different configurations work well on different MOPs.

Note that some recent efforts have been made to adjust the hyperparameters of MOEA/D by heuristic rules [31], multi-armed bandit [21], and RL [43]. However, they all focus on a single type of hyperparameter, and also very few of them work on the dynamic algorithm configuration well.

## 4.2 MA-DAC for MOEA/D

In this section, we show how to apply MA-DAC to learn the configuration policy of MOEA/D and also introduce the resulting environment MaMo. Each generation in the evolutionary process of MOEA/D is one step in the MaMo.

The state of MaMo can be divided into three parts.

1. To describe the general properties of the optimization problems, the first part is the features of the problem instance, including the number of variables and objectives.

2. To emphasize the general information of the algorithm, the second part includes the features of the optimization process, i.e., how much computational budget has been used and how many steps of the algorithm have not made any progress.

3. To show the properties of the population and that how the population is evolved, we use several indicators, i.e., hypervolume[2] [53], ratio of non-dominated solutions in the population and average distance of the solutions, in the third part. For each indicator, we also use the gap between the current value and the value corresponding to the last population to

---

[1]For two solutions $x$ and $y$, $x$ is called to dominate $y$, if $x$ is not worse than $y$ on all objectives and better than $y$ on at least one objective.

[2]Hypervolume is a quality indicator that can reflect both convergence and diversity of a solution set [22].

reflect the immediate evolutionary progress. Besides, we use statistic metrics (the mean and standard deviation) of the indicators in several recent steps and all steps from the beginning to measure the short and long histories of the optimization, respectively.

To facilitate the generalization of the learned MA-DAC policy, we have pre-processed some state features. More details about state are provided in Appendix A.2 due to space limitation.

We consider four heterogeneous types of configuration hyperparameters in MOEA/D as the actions of different agents of MA-DAC.

1. Weights. In MOEA/D, weights are used to maintain the diversity of the converted single-objective sub-problems [31]. The action space is discrete with two dimensions, i.e., adjusting (T) and not adjusting (N) the weights. Furthermore, we limit the frequency of adjustment, because too frequent adjustment will lead to drastic changes in the sub-problems and is detrimental to the optimization process.

2. Neighborhood size. The neighborhood size is to control the distance between solutions in mating selection; a small size helps the search exploit the local area while a large size helps the search explore a wide objective space [47]. We discretize the action space into four dimensions. That is, the neighborhood size can be 15, 20, 25 or 30.

3. Types of the reproduction operators. We consider four types of differential evolution (DE) operators. Each type has different search ability [21, 36].

4. Parameters of reproduction operators. The parameters (e.g., scaling factor) of the reproduction operators in MOEA/D significantly affect the algorithm's performance [39], and the action space has four discrete dimensions.

The detailed definitions of the actions are given in Appendix A.3 due to space limitation.

We use the MOPs with similar properties from the well-known MOP benchmarks DTLZ [9] and WFG [16] as the instance set of MaMo. These MOPs can have different number of objectives, and those with the same number of objectives can be seen as inner classes because the number of objectives considerably affects the properties and difficulty level of MOPs. We use the inverted generational distance (IGD) [4] as the metric in Eq. (3), resulting in the reward function.

To learn the policy, we use a classical MARL algorithm named VDN [40], which is widely used in cooperative multi-agent systems. VDN learns to decompose the team value function into agent-wise value functions, alleviating the exponential growth of the action space. It follows the Individual-Global-Max (IGM) principle [37], i.e., the consistency between joint and local greedy action selections by the joint value function $Q_{\text{tot}}(s, \boldsymbol{a})$ and individual value functions $\left[Q_j(s, a^{(j)})\right]_{j=1}^n$. All parameters in MA-DAC are updated using the standard TD loss from the global Q-value $Q_{\text{tot}}$, which follows additivity to factorize the global value function, i.e., $Q_{\text{tot}}(s, \boldsymbol{a}) = \sum_{j=1}^n Q_j\left(s, a^{(j)}\right)$. In the testing phase, each agent acts greedily with respect to its $Q_j$.

## 4.3 Comparison among MARL benchmarks

There are many benchmarks [30, 42] emerged in recent years. In order to clearly show the characteristics of MaMo, we compare it with some MARL benchmarks, as shown in Table 1. Heterogeneity refers to agents having different action spaces or skills [45]. For example, in MaMo, there are four entirely different action spaces, i.e., weights, neighborhood size, types of the reproduction operators and their parameters. Stochasticity means that performing the same action in the same state may lead to a different next state. The high stochasticity of MaMo comes from the randomness of MOEA/D.

The primary benchmarks are designed to investigate different aspects of multi-agent systems. For example, Matrix Games [6] and Multi-Agent Particle Environments (MPE) [25] are popular and classical testbeds, to investigate cooperative or competitive behaviors in small scale settings (with 2–3 agents). MAgent [52] is used to test the scalability of MARL methods, where the number of agents is up to 1000. SMAC [34] has attracted wide attention in recent years to test the coordination ability in cooperative MARL [11, 49, 30]. For the first time, Active Voltage Control [46] creates an exciting yet challenging real-world scenario for the application of MARL. To the best of our knowledge, none of the known benchmarks has focused on highly heterogeneous and stochastic scenarios in MARL. We hope our new MaMo benchmark can offer a good supplement that could benefit the MARL community.

Table 1: Overview of MARL benchmarks and their properties.

| Benchmark | Heterogeneous | #agents | Stochastic | Application scenarios |
|---|---|---|---|---|
| Matrix Games [6] | × | 2 | Low | Game |
| MPE [25] | × | 2-3 | Low | Game |
| MAgent [52] | × | 2-1000 | Low | Game |
| SMAC [34] | ✓ | 2-30 | Low | Game |
| Active Voltage Control [46] | × | 3-38 | Low | Control |
| MaMo (Ours) | ✓ | 2-4 | High | Optimization |

## 5 Experiments

To examine the effectiveness of MA-DAC, we conduct experiments using MaMo. We investigate the following research questions (RQs). RQ1: How does MA-DAC perform compared with the baseline and other tuning algorithms? RQ2: How is the generalization ability of MA-DAC? RQ3: How do the different parts of MA-DAC affect the performance? We introduce the experimental settings in Section 5.1, and investigate the above RQs in Section 5.2.

### 5.1 Experimental settings

We consider several well-established multi-objective test functions (DTLZ2, DTLZ4 and WFG4-WFG9) with 3, 5 and 7 objectives following the practice in Bezerra et al. [2] to examine our method. In all of the following experiments, several arbitrary functions (here DTLZ2, WFG4 and WFG6) from the function set are used as the training set, and all the other functions are considered as the testing set. Note that "train" and "test" in the tables mean that the comparison is based on the training (i.e., DTLZ2, WFG4 and WFG6) and testing (i.e., DTLZ4, WFG5, WFG7, WFG8 and WFG9) problems, respectively. IGD [4] is considered as the metric to measure the performance of the algorithms. The mean and standard deviation of the IGD values obtained by each algorithm on each MOP for 30 independent runs are reported. We apply the Wilcoxon rank-sum test with significance level 0.05. For a fair comparison, other parameters (such as population size and computational budget) are set to be the same across all the compared algorithms. More details about the settings can be found in Appendix B.1 due to space limitation. Our code is available at https://github.com/lamda-bbo/madac.

### 5.2 Experimental results

**RQ1: How does MA-DAC perform compared with the baseline and other tuning algorithms?**
We compare the MA-DAC policy with the original MOEA/D [51], DQN [28] and MA-UCB [15]. DQN is a single-agent RL algorithm that shares the same state, reward, and transition as our MA-DAC but has a different action space that is the concatenation of the four types of hyperparameters. DQN can be seen as an instantiation of DAC in MaMo. MA-UCB is a simplified algorithm that uses four UCB agents to adjust the four types of hyperparameters. For MA-DAC and DQN, we perform the training and testing on the problems with the number of objectives.

The results are shown in Table 2. As can be seen in the table, MA-DAC is significantly superior to the other algorithms on almost all the 24 problems. Furthermore, MA-DAC (which is trained only on DTLZ2, WFG4 and WFG6) shows excellent performance on previously unseen problems during the training process, demonstrating its good generalization ability. DQN policy is significantly worse than MA-DAC on all the problems, which is consistent with the previous observation that DQN performs poorly in MARL tasks [13]. DQN suffers from the exponentially increasing action space with the number of agents [29, 50], while MA-DAC effectively decomposes the action space, making each agent much easier to find their own near-optimal policy and then leading to a good joint policy. The superiority of MA-DAC over MA-UCB also demonstrates the necessity of learning configuration policy by an MARL algorithm.

We also compare our algorithm with two adaptive MOEA/D algorithms, i.e., MOEA/D-FRRMAB [21] and MOEA/D-AWA [31], which adjust reproduction operator types and weights based on MAB and heuristic rules, respectively. The results clearly show the superior performance of MA-DAC, and are provided in Appendix B.3 and Appendix B.4 due to space limitation.

Table 2: IGD values obtained by MOEA/D, DQN, MA-UCB and MA-DAC on different problems. Each result consists of the mean and standard deviation of 30 runs. The best mean value on each problem is highlighted in **bold**. The symbols '+', '−' and '≈' indicate that the result is significantly superior to, inferior to, and almost equivalent to MA-DAC, respectively, according to the Wilcoxon rank-sum test with significance level 0.05.

| Problem | $M$ | MOEA/D | DQN | MA-UCB | MA-DAC |
|---|---|---|---|---|---|
| DTLZ2 | 3 | 4.605E-02 (3.54E-04) − | 4.628E-02 (2.96E-04) − | 4.671E-02 (3.70E-04) − | **3.807E-02** (5.05E-04) |
|  | 5 | 3.006E-01 (1.55E-03) − | 3.016E-01 (1.34E-03) − | 3.041E-01 (1.69E-03) − | **2.442E-01** (1.26E-02) |
|  | 7 | 4.455E-01 (1.41E-02) − | 4.671E-01 (1.15E-02) − | 4.826E-01 (9.59E-03) − | **3.944E-01** (1.17E-02) |
| WFG4 | 3 | 5.761E-02 (5.41E-04) − | 6.920E-02 (1.20E-03) − | 7.165E-02 (1.83E-03) − | **5.200E-02** (1.19E-03) |
|  | 5 | 3.442E-01 (1.21E-02) − | 2.810E-01 (6.86E-03) − | 2.859E-01 (6.77E-03) − | **1.868E-01** (2.81E-03) |
|  | 7 | 4.529E-01 (1.79E-02) − | 3.725E-01 (1.14E-02) − | 3.868E-01 (1.54E-02) − | **3.033E-01** (3.66E-03) |
| WFG6 | 3 | 6.938E-02 (5.50E-03) − | 6.834E-02 (1.78E-02) − | 6.601E-02 (1.00E-02) − | **4.831E-02** (8.95E-03) |
|  | 5 | 3.518E-01 (2.82E-03) − | 3.160E-01 (2.40E-02) − | 3.359E-01 (1.47E-02) − | **1.942E-01** (6.90E-03) |
|  | 7 | 4.869E-01 (3.03E-02) − | 4.322E-01 (2.95E-02) − | 4.389E-01 (3.41E-02) − | **3.112E-01** (4.93E-03) |
| Train: +/−/≈ |  | 0/9/0 | 0/9/0 | 0/9/0 |  |
| DTLZ4 | 3 | 6.231E-02 (8.85E-02) ≈ | **5.590E-02** (5.77E-03) − | 6.011E-02 (5.08E-03) − | 6.700E-02 (6.14E-02) |
|  | 5 | 3.133E-01 (4.45E-02) ≈ | 3.457E-01 (1.61E-02) − | 3.492E-01 (1.69E-02) − | **2.995E-01** (2.10E-02) |
|  | 7 | 4.374E-01 (2.57E-02) − | 4.552E-01 (1.47E-02) − | 4.756E-01 (2.01E-02) − | **4.182E-01** (1.21E-02) |
| WFG5 | 3 | 6.327E-02 (1.10E-03) − | 6.212E-02 (5.54E-04) − | 6.118E-02 (7.03E-04) − | **4.730E-02** (7.89E-04) |
|  | 5 | 3.350E-01 (9.77E-03) − | 3.077E-01 (6.36E-03) − | 3.036E-01 (8.83E-03) − | **1.811E-01** (3.02E-03) |
|  | 7 | 4.101E-01 (2.08E-02) − | 4.996E-01 (1.32E-02) − | 5.024E-01 (1.38E-02) − | **3.206E-01** (8.04E-03) |
| WFG7 | 3 | 5.811E-02 (6.31E-04) − | 5.930E-02 (7.32E-04) − | 6.014E-02 (7.11E-04) − | **4.066E-02** (5.31E-04) |
|  | 5 | 3.572E-01 (5.47E-03) − | 2.993E-01 (1.43E-02) − | 3.207E-01 (1.71E-02) − | **1.858E-01** (2.12E-03) |
|  | 7 | 5.236E-01 (2.19E-02) − | 4.576E-01 (2.38E-02) − | 4.879E-01 (2.75E-02) − | **3.258E-01** (1.25E-02) |
| WFG8 | 3 | 8.646E-02 (3.44E-03) − | 9.280E-02 (1.06E-03) − | 9.612E-02 (1.48E-03) − | **7.901E-02** (1.19E-03) |
|  | 5 | 4.258E-01 (8.42E-03) − | 3.969E-01 (1.26E-02) − | 3.956E-01 (1.32E-02) − | **2.479E-01** (7.20E-03) |
|  | 7 | 5.816E-01 (1.30E-02) − | 5.575E-01 (1.39E-02) − | 5.642E-01 (1.38E-02) − | **4.127E-01** (5.93E-03) |
| WFG9 | 3 | 5.817E-02 (1.24E-03) − | 5.628E-02 (7.29E-04) − | 7.953E-02 (2.45E-02) − | **4.159E-02** (6.10E-04) |
|  | 5 | 3.633E-01 (1.20E-02) − | 3.258E-01 (1.61E-02) − | 3.396E-01 (1.55E-02) − | **1.832E-01** (7.10E-03) |
|  | 7 | 5.538E-01 (2.63E-02) − | 5.115E-01 (2.15E-02) − | 5.227E-01 (1.79E-02) − | **3.278E-01** (7.21E-03) |
| Test: +/−/≈ |  | 0/13/2 | 0/15/0 | 0/15/0 |  |

**RQ2: How is the generalization ability of MA-DAC?** We compare MA-DAC policies trained from different training sets to test the generalization ability of MA-DAC policies. MA-DAC (M) denotes that the policy is trained on the problems (i.e., DTLZ2, WFG4 and WFG6) with all 3, 5 and 7 objectives. Meanwhile, MA-DAC (3), (5) and (7) denote that the policies are trained on the problems (i.e., DTLZ2, WFG4 and WFG6) with 3, 5 and 7 objectives, respectively.

As shown in Table 3, it is unsurprising that the results of average rank show that the MA-DAC (3), (5) and (7) policies have excellent performance on the problems with 3, 5 and 7 objectives, respectively. For example, MA-DAC (7) has the best average rank (i.e., 1.0) on the test problems with 7 objectives, but has the worst average rank (i.e., 4.0) on the problems with 3 objectives. To obtain a more robust policy, we mix the problems with different number of objectives as the training set, resulting in the MA-DAC (M). As can be seen in Table 3, MA-DAC (M) demonstrates its robustness. Among all the policies, MA-DAC (M) takes the first, third, and second places in terms of its performance in the three types of problems, respectively. Lastly, the last row of the table shows the out-performance of these MA-DAC policies compared with other policies – the IGD values of all the four MA-DAC policies perform significantly better than the best result obtained by the three peer algorithms MOEA/D, DQN and MA-UCB in Table 2.

**RQ3: How do the different parts of MA-DAC affect the performance?** We conduct ablation studies to show the importance of different types of hyperparameters, and the results are given in Table 4. We use MA-DAC (M) w/o $i$ to denote the MA-DAC policy that does not include the $i$-th agent, and we use a reasonable setting as the default for each ablated agent; the detailed settings are provided in Appendix B.1. That is, MA-DAC (M) w/o 1, 2, 3 and 4 represent MA-DAC (M) without tuning weights, neighborhood size, types of reproduction operators, and parameters of reproduction operators, respectively.

As can be seen in Table 4. MA-DAC (M) outperforms all the ablations, demonstrating the importance of tuning every type of hyperparameter. Note that the column of MA-DAC (M) in Table 4 is just as

Table 3: IGD values obtained by MA-DAC (M), MA-DAC (3), MA-DAC (5) and MA-DAC (7) on different problems. Each result consists of the mean and standard deviation of 30 runs. The best mean value on each problem is highlighted in **bold**. The symbols '+', '−' and '≈' indicate that the result is significantly superior to, inferior to, and almost equivalent to the best value except for MA-DAC in Table 2, respectively, according to the Wilcoxon rank-sum test with significance level 0.05.

| Problem | $M$ | MA-DAC (M) | MA-DAC (3) | MA-DAC (5) | MA-DAC (7) |
|---|---|---|---|---|---|
| DTLZ2 | 3 | 3.839E-02 (5.35E-04) + | **3.807E-02** (5.05E-04) + | 3.830E-02 (7.24E-04) + | 3.837E-02 (5.69E-04) + |
|  | 5 | 2.468E-01 (7.55E-03) + | 2.472E-01 (1.56E-02) + | **2.442E-01** (1.26E-02) + | 2.569E-01 (1.39E-02) + |
|  | 7 | 3.921E-01 (8.84E-03) + | **3.880E-01** (1.02E-02) + | 4.081E-01 (1.52E-02) + | 3.944E-01 (1.17E-02) + |
| WFG4 | 3 | 5.220E-02 (9.83E-04) + | **5.200E-02** (1.19E-03) + | 5.236E-02 (1.10E-03) + | 5.302E-02 (9.78E-04) + |
|  | 5 | **1.850E-01** (3.14E-03) + | 1.867E-01 (3.01E-03) + | 1.868E-01 (2.81E-03) + | 1.853E-01 (2.67E-03) + |
|  | 7 | 3.091E-01 (5.80E-03) + | 3.104E-01 (7.14E-03) + | 3.100E-01 (5.89E-03) + | **3.033E-01** (3.66E-03) + |
| WFG6 | 3 | 5.078E-02 (1.20E-02) + | 4.831E-02 (8.95E-03) + | **4.599E-02** (9.48E-03) + | 5.206E-02 (1.64E-02) + |
|  | 5 | 1.971E-01 (6.40E-03) + | 2.003E-01 (6.26E-03) + | **1.942E-01** (6.90E-03) + | 1.957E-01 (6.67E-03) + |
|  | 7 | 3.114E-01 (5.08E-03) + | 3.242E-01 (9.24E-03) + | 3.129E-01 (5.71E-03) + | **3.112E-01** (4.93E-03) + |
| DTLZ4 | 3 | **6.171E-02** (3.67E-02) + | 6.700E-02 (6.14E-02) + | 6.618E-02 (4.62E-02) + | 8.088E-02 (7.12E-02) + |
|  | 5 | 3.044E-01 (1.66E-02) ≈ | **2.974E-01** (1.94E-02) + | 2.995E-01 (2.10E-02) ≈ | 3.036E-01 (1.69E-02) ≈ |
|  | 7 | 4.271E-01 (1.45E-02) + | 4.313E-01 (1.39E-02) ≈ | 4.327E-01 (2.15E-02) ≈ | **4.182E-01** (1.21E-02) + |
| WFG5 | 3 | **4.721E-02** (7.15E-04) + | 4.730E-02 (7.89E-04) + | 4.733E-02 (8.10E-04) + | 4.746E-02 (5.90E-04) + |
|  | 5 | 1.811E-01 (2.59E-03) + | 1.817E-01 (2.96E-03) + | 1.811E-01 (3.02E-03) + | **1.808E-01** (2.83E-03) + |
|  | 7 | 3.256E-01 (5.49E-03) + | 3.266E-01 (8.98E-03) + | 3.263E-01 (9.73E-03) + | **3.206E-01** (8.04E-03) + |
| WFG7 | 3 | 4.076E-02 (5.33E-04) + | **4.066E-02** (5.31E-04) + | 4.077E-02 (5.12E-04) + | 4.124E-02 (4.98E-04) + |
|  | 5 | 1.839E-01 (2.38E-03) + | 1.881E-01 (3.70E-03) + | 1.858E-01 (2.12E-03) + | **1.836E-01** (2.21E-03) + |
|  | 7 | 3.368E-01 (1.54E-02) + | 3.461E-01 (1.97E-02) + | 3.390E-01 (1.38E-02) + | **3.258E-01** (1.25E-02) + |
| WFG8 | 3 | **7.828E-02** (1.46E-03) + | 7.901E-02 (1.19E-03) + | 7.921E-02 (1.36E-03) + | 7.944E-02 (1.30E-03) + |
|  | 5 | 2.506E-01 (1.11E-02) + | 2.653E-01 (1.51E-02) + | **2.479E-01** (7.20E-03) + | 2.532E-01 (9.28E-03) + |
|  | 7 | 4.303E-01 (1.49E-02) + | 4.364E-01 (1.38E-02) + | 4.242E-01 (9.08E-03) + | **4.127E-01** (5.93E-03) + |
| WFG9 | 3 | 4.324E-02 (7.07E-04) + | **4.159E-02** (6.10E-04) + | 4.359E-02 (1.00E-02) + | 6.415E-02 (2.64E-02) + |
|  | 5 | 1.858E-01 (7.63E-03) + | **1.814E-01** (4.59E-03) + | 1.832E-01 (7.10E-03) + | 1.918E-01 (1.13E-02) + |
|  | 7 | 3.328E-01 (1.02E-02) + | 3.298E-01 (1.03E-02) + | 3.307E-01 (1.37E-02) + | **3.278E-01** (7.21E-03) + |
| Test: average rank | 3 | 1.4 | 1.8 | 2.8 | 4.0 |
|  | 5 | 2.6 | 2.8 | 2.2 | 2.4 |
|  | 7 | 2.6 | 3.4 | 3.0 | 1.0 |
| Test: +/−/≈ | 3 | 5/0/0 | 5/0/0 | 5/0/0 | 5/0/0 |
|  | 5 | 4/0/1 | 5/0/0 | 4/0/1 | 4/0/1 |
|  | 7 | 5/0/0 | 4/0/1 | 4/0/1 | 5/0/0 |

same as that in Table 3. On the other hand, we notice that the importance of hyperparameters varies. For example, adaptive weights are in general more important as the performance of MA-DAC (M) w/o 1 drops significantly.

**Further studies** Due to the space limitation, more experiments on MaMo and DACBench [10] are provided in Appendix B and Appendix C, respectively.

- To show the effectiveness of the proposed triangle-based reward function, we compare it with different reward functions in Appendix B.2.

- We give a detailed analysis of the reproduction operators and the adaptive weights in Appendix B.3 and Appendix B.4, respectively.

- To show the optimization process of each method, we plot the curves of IGD value of different methods (i.e., MOEA/D, MOEA/D-FRRMAB, MOEA/D-AWA, DQN, MA-UCB and MA-DAC) in Appendix B.5.

- To improve the compared baseline, we use DQN to dynamically adjust each type of hyperparameter of MOEA/D in Appendix B.6.

- We use two more MARL algorithms as the implementations of policy networks, i.e., Independent Q-Learning (IQL) [41] and QMIX [32] in Appendix B.7.

- We also conduct experiments on Sigmoid from DACBench [10] in Appendix C, which can generate instance sets with a wide range of difficulties. We conduct experiments on the $5D$-Sigmoid and $10D$-Sigmoid (i.e., there are 5 and 10 agents in MA-DAC, respectively) with action space size 3 (i.e., each agent has a 3-dimensional discrete action space). The experimental results show the versatility and scalability of MA-DAC.

Table 4: IGD values obtained by MA-DAC (M) w/o 1, MA-DAC (M) w/o 2, MA-DAC (M) w/o 3 and MA-DAC (M) w/o 4 on different problems. Each result consists of the mean and standard deviation of 30 runs. The best mean value on each problem is highlighted in **bold**. The symbols '+', '−' and '≈' indicate that the result is significantly superior to, inferior to, and almost equivalent to MA-DAC (M), respectively, according to the Wilcoxon rank-sum test with significance level 0.05.

| Problem | $M$ | MA-DAC (M) w/o 1 | MA-DAC (M) w/o 2 | MA-DAC (M) w/o 3 | MA-DAC (M) w/o 4 | MA-DAC (M) |
|---|---|---|---|---|---|---|
| DTLZ2 | 3 | 4.656E-02 (3.80E-04) − | 3.914E-02 (8.43E-04) − | 3.935E-02 (6.72E-04) − | 3.919E-02 (5.91E-04) − | **3.839E-02** (5.35E-04) |
|  | 5 | 3.086E-01 (7.24E-03) − | 2.619E-01 (8.99E-03) − | 2.503E-01 (1.30E-02) − | **2.433E-01** (1.59E-02) ≈ | 2.468E-01 (7.55E-03) |
|  | 7 | 4.970E-01 (1.26E-02) − | 4.067E-01 (1.20E-02) − | 4.003E-01 (1.19E-02) − | 4.228E-01 (1.25E-02) − | **3.921E-01** (8.84E-03) |
| WFG4 | 3 | 7.222E-02 (1.93E-03) − | 5.484E-02 (1.01E-03) − | 5.410E-02 (8.85E-04) − | 5.410E-02 (8.85E-04) − | **5.220E-02** (9.83E-04) |
|  | 5 | 2.868E-01 (1.01E-02) − | 1.879E-01 (3.76E-03) ≈ | **1.845E-01** (2.17E-03) + | 1.846E-01 (2.39E-03) + | 1.850E-01 (3.14E-03) |
|  | 7 | 3.758E-01 (1.33E-02) − | 3.102E-01 (6.34E-03) − | **3.020E-01** (3.99E-03) ≈ | 3.032E-01 (4.32E-03) − | 3.091E-01 (5.80E-03) |
| WFG6 | 3 | 6.864E-02 (8.14E-03) − | 5.338E-02 (1.37E-02) − | 6.543E-02 (1.69E-02) − | 6.067E-02 (2.11E-02) ≈ | **5.078E-02** (1.20E-02) |
|  | 5 | 3.480E-01 (1.34E-02) − | 2.005E-01 (5.21E-03) − | 1.996E-01 (6.51E-03) − | 1.979E-01 (6.76E-03) − | **1.971E-01** (6.40E-03) |
|  | 7 | 4.784E-01 (3.37E-02) − | 3.147E-01 (5.85E-03) − | 3.162E-01 (6.15E-03) − | 3.147E-01 (5.73E-03) − | **3.114E-01** (5.08E-03) |
| Train: +/−/≈ |  | 0/9/0 | 0/8/1 | 1/7/1 | 1/5/3 |  |
| DTLZ4 | 3 | 6.463E-02 (3.85E-02) − | 6.242E-02 (4.07E-02) − | **4.496E-02** (2.45E-03) + | 4.496E-02 (2.45E-03) + | 6.171E-02 (3.67E-02) |
|  | 5 | 3.497E-01 (1.41E-02) − | 3.061E-01 (2.12E-02) ≈ | 3.054E-01 (1.55E-02) ≈ | 3.130E-01 (1.81E-02) − | **3.044E-01** (1.66E-02) |
|  | 7 | 4.853E-01 (1.82E-02) − | 4.275E-01 (1.90E-02) − | **4.208E-01** (1.52E-02) ≈ | 4.289E-01 (2.07E-02) − | 4.271E-01 (1.45E-02) |
| WFG5 | 3 | 6.189E-02 (7.40E-04) − | 4.827E-02 (6.31E-04) − | 4.766E-02 (5.72E-04) ≈ | 4.766E-02 (5.72E-04) ≈ | **4.721E-02** (7.15E-04) |
|  | 5 | 3.202E-01 (9.77E-03) − | 1.821E-01 (2.73E-03) ≈ | 1.835E-01 (2.90E-03) − | 1.822E-01 (2.31E-03) − | **1.811E-01** (2.59E-03) |
|  | 7 | 4.948E-01 (1.47E-02) − | 3.290E-01 (8.87E-03) − | 3.310E-01 (7.70E-03) − | 3.261E-01 (9.26E-03) − | **3.256E-01** (5.49E-03) |
| WFG7 | 3 | 6.004E-02 (9.45E-04) − | 4.250E-02 (5.82E-04) − | 4.150E-02 (6.60E-04) − | 4.150E-02 (6.60E-04) − | **4.076E-02** (5.33E-04) |
|  | 5 | 3.402E-01 (2.49E-02) − | 1.873E-01 (3.67E-03) ≈ | **1.826E-01** (2.60E-03) + | 1.847E-01 (2.80E-03) ≈ | 1.839E-01 (2.38E-03) |
|  | 7 | 4.877E-01 (5.70E-02) − | 3.393E-01 (1.23E-02) − | 3.373E-01 (1.16E-02) − | 3.377E-01 (1.59E-02) − | **3.368E-01** (1.54E-02) |
| WFG8 | 3 | 9.661E-02 (1.87E-03) − | 8.374E-02 (1.70E-03) − | 8.029E-02 (1.29E-03) − | 8.029E-02 (1.29E-03) − | **7.828E-02** (1.46E-03) |
|  | 5 | 4.119E-01 (1.30E-02) − | 2.695E-01 (1.49E-02) − | 2.571E-01 (1.09E-02) − | 2.632E-01 (9.96E-03) − | **2.506E-01** (1.11E-02) |
|  | 7 | 5.830E-01 (1.59E-02) − | 4.345E-01 (9.27E-03) − | **4.260E-01** (8.71E-03) − | 4.322E-01 (1.12E-02) − | 4.303E-01 (1.49E-02) |
| WFG9 | 3 | 5.894E-02 (9.24E-04) − | **4.321E-02** (7.50E-04) − | 4.650E-02 (1.40E-02) − | 4.650E-02 (1.40E-02) − | 4.324E-02 (7.07E-04) |
|  | 5 | 3.148E-01 (2.06E-02) − | 1.875E-01 (5.14E-03) − | 1.941E-01 (6.45E-03) − | 1.865E-01 (9.02E-03) − | **1.858E-01** (7.63E-03) |
|  | 7 | 5.069E-01 (2.53E-02) − | 3.433E-01 (1.35E-02) − | 3.402E-01 (1.00E-02) − | 3.368E-01 (1.05E-02) − | **3.328E-01** (1.02E-02) |
| Test: +/−/≈ |  | 0/15/0 | 0/12/3 | 2/10/3 | 1/12/2 |  |

## 6 Conclusion

This paper considers the dynamic configuration of algorithms with multiple types of hyperparameters. We propose MA-DAC to solve it, where one agent works to handle one type of configuration hyperparameter. Experimental results show that MA-DAC works well and has good generalization ability. The instantiation of configuring MOEA/D forms the benchmark MaMo for MARL, with the hope of facilitating the application of MARL.

Considering the superior performance of MA-DAC in empirical studies, an interesting future work is to perform theoretical analysis, to better understand why MA-DAC can work. Particularly, many MARL algorithms follow the IGM principle that assumes the global Q is factorizable, while it is not yet clear whether DAC problems are (approximately) factorizable. In addition, we will try to propose better contextual MMDP formulation as well as better cooperative MARL algorithms, based on the heterogeneity and stochasticity of MA-DAC. It is also interesting to include more real-world optimization problems into MaMo.

## Acknowledgements

We thank the reviewers for their insightful and valuable comments. We thank Chenghe Wang and Hao Yin for providing helpful comments. This work was supported by the National Key Research and Development Program of China (2020AAA0107200), the NSFC (62022039, 62276124, 61876119), the Fundamental Research Funds for the Central Universities (0221-14380009, 0221-14380014), and the program B for Outstanding Ph.D. candidate of Nanjing University.

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
