# OpenReview forum: "Multi-agent Dynamic Algorithm Configuration"
_NeurIPS.cc/2022/Conference — NeurIPS 2022 Accept_

### Official Review · Reviewer_55wR · 2022-07-01

**Rating:** 8
**Confidence:** 4
**Soundness:** 3 good
**Presentation:** 3 good
**Contribution:** 4 excellent

**Summary:**

This paper addresses the problem of adjusting multiple heterogeneous hyper-parameters of a complex algorithm dynamically, and proposes a general framework, Multi-Agent Dynamic Algorithm Configuration (MA-DAC). The authors gave a concrete application of tuning the hyper-parameters of a popular evolutionary algorithm MOEA/D for solving multi-objective optimisation problems, and the experimental results on benchmark multi-objective problems show the effectiveness of MA-DAC.

**Questions:**

- I did not see any discussion on the scalability of MA-DAC. The application of MaMo involves four heterogeneous hyper-parameters. If there are more hyper-parameters, how about the runtime complexity of MA-DAC?

- For RQ2 in the experiments, “MA-DAC (M) … trained on the problems with all 3, 5, and 7 objectives” What are the training problems? Still DTLZ2, WFG4 and WFG6?

- If I have not misunderstood, MA-DAC (M) in Table 3 is just MA-DAC in Table 2. If so, why are the results slightly different?

- The states in Table S1 defined for MOEA/D seem to be reasonable and comprehensive. Can they be directly applied when tuning the hyper-parameters of other MOEAs (e.g., NSGA-II and SPEA)? I’d like to see some discussion.

- lines 109-110 in the appendix, what does $x$ mean in OP3 and OP4?

- In the experiments, you used the popular benchmark multi-objective problems, DTLZ and WFG. I understand these are standard test problems. But if you can add some real-world problems, the contribution will be more significant.


**Limitations:**

YES

**Strengths And Weaknesses:**

DAC (Biedenkapp et al., ECAI’20) has been an effective framework for dynamically adjusting a single type of hyper-parameters of algorithms, while this paper tries to address the more challenging problem, that is adjusting multiple heterogeneous hyper-parameters simultaneously. The authors modeled this problem as a contextual multi-agent MDP where one agent corresponds to one type of hyper-parameter, and applied multi-agent RL algorithms to solve it.

As far as I can tell, this is the first work to address the dynamic tuning of multiple heterogeneous types of hyper-parameters. The idea of multi-agent MDP modeling is natural, and also reasonable. The framework named MA-DAC is general. I like the authors gave guidance on the setting of elements of multi-agent MDP, which makes the MA-DAC framework easier to be applied. The triangle-shaped reward function encouraging long-term rewards is novel, the effectiveness of which the authors have also empirically validated. Very interesting work!

To show the effectiveness of MA-DAC, the authors considered the MOEA/D (Zhang & Li, TEvC’07), which is a popular evolutionary algorithm for solving multi-objective optimisation problems. They applied MA-DAC to dynamically adjust the hyper-parameters of MOEA/D, including weights, neighborhood size, types and parameters of crossover operator. This is a meaningful and practical application, and also constitutes a difficult benchmark (named MaMo in the paper) for MARL algorithms, which can be independent of interest for the MARL community.

In the experiments, the authors selected the DTLZ and WFG benchmark problems with various number of objectives, three of which are used for training and the others are for testing. They compared MA-DAC with DAC (using one agent to adjust the combination of the four heterogeneous hyper-parameters together), MA-UCB (using four UCB agents to adjust the four heterogeneous hyper-parameters, respectively) and two SOTA adaptive MOEA/D algorithms. Their experimental results clearly show the superiority of MA-DAC (MA-DAC is significantly better in most cases). Especially, the generalization ability of MA-DAC is impressive to me. MA-DAC trained using problems with a single number of objectives can even be better than the compared algorithms trained using problems with various number of objectives.

I have read the appendix as well. The experiments are described in details, and I believe they are reproducible. The codes are also provided.

In my opinion, this is a very good work. The authors studied the challenging problem of dynamically tuning multiple heterogeneous types of hyper-parameters for the first time, proposed a simple, yet novel and general multi-agent RL solution, gave a practical application of tuning the hyper-parameters of MOEA/D, and showed the excellent performance by comprehensive experiments.

I believe MA-DAC would be a supplementary for DAC, as well as a good resource to the AutoML community, and will bring some influence.

The paper is also well written and easy to follow. Minor issues:

- line 17, “problems classes” -> “problem classes”
- line 31, “the tuning learning rate” -> “the learning rate tuning”
- line 72, “the the” -> “the”
- Tables 3, 4, S3, S4, S5, S6, caption, “The best mean value of each problem” -> “The best mean value on each problem”
- Table 4, caption, it would be better to clearly say “MA-DAC (M) in Table 3”, which is easier to be understood.
- line 401, “nature” -> “Nature”
- line 40 in the appendix, “lines 4-9” -> “lines 4-10”
- line 43 in the appendix, “reproduction” -> “crossover”
- line 136 in the appendix, “at step $t$” -> “by step $t$”?
- Table S3, caption, “the MA-DAC” -> “MA-DAC”

For my major concerns, please see the questions.

---

> ### Author Response · Authors · 2022-08-02
> **Response to Reviewer 55wR**
>
>
> Thank you very much for carefully reviewing our paper and providing constructive comments and suggestions. We
> are very glad that you appreciate our work. Below please find our response.
>
> ## Q1 Scalability of MA-DAC
> The run-time complexity and scalability of MA-DAC relies on the MARL algorithms used. Let us take the employed MARL algorithm VDN as an example. VDN trains individual agents with a linear decomposition network architecture, which learns to decompose the total reward into the agent-wise reward. Thus, the run-time complexity of MA-DAC increases linearly with the number of agents. Improving the scalability by techniques such as grouping would be of great interest in MARL community, which have been proved efficient in some recent works [1-3]. When the scale of the task is very large (e.g., 100+), we can try these techniques. But note that most algorithms do not have so many hyperparameters to tune. Thanks to your comments, we have discussed this and conducted experiment on another well-known benchmark, i.e., DACBench [4], in Appendix C of the revised paper.
>
> [1] VAST: Value Function Factorization with Variable Agent Sub-Teams. NeurIPS, 2021.
>
> [2] Scaling Multi-Agent Reinforcement Learning with Selective Parameter Sharing. ICML, 2021.
>
> [3] Concentration Network for Reinforcement Learning of Large-Scale Multi-Agent Systems. AAAI, 2022.
>
> [4] DACBench: A Benchmark Library for Dynamic Algorithm Configuration. IJCAI, 2021.
>
> ## Q2 What are the training problems in RQ2? Still DTLZ2, WFG4 and WFG6?
> Yes. We use DTLZ2, WFG4 and WFG6 as our training problems in all the experiments on MaMo.
>
> ## Q3 MA-DAC (M) in Table 3 is just MA-DAC in Table 2?
>
> MA-DAC(M) in Table 3 is not MA-DAC in Table 2. In Table 3, MA-DAC(M) is trained on 9 types of functions, i.e., DTLZ2, WFG4, and WFG6 with 3, 5, and 7 objectives, and tested on all the other functions. In Table 2, the functions with different number of objectives are treated separately; e.g., MA-DAC is trained on the functions DTLZ2, WFG4, and WFG6 with 3 objectives, and tested on the other functions with the same number of objectives.
>
> ## Q4 Discussion of other MOEA algorithms
>
> We think that the states defined for MOEA/D can be directly used to tune the hyper-parameters of other MOEAs. Compared with the Pareto-based algorithms (e.g., NSGA-II and SPEA), decomposition-based algorithms (e.g., MOEA/D) have more types of hyper-parameters (e.g., weights and neighborhood size). Thus, we focused on the dynamic configuration of MOEA/D in this paper. We will try to apply MA-DAC to tune the hyper-parameters of other MOEAs in the future. Thanks for your suggestions.
>
> ## Q5 What does $\boldsymbol{x}$ mean in OP3 and OP4?
>
> $\boldsymbol{x}$ should be $\boldsymbol{x}^{(i)}$. We are sorry for the typo.
>
> ## Q6 Real-world applications
> We agree that adding more real-world applications to the MaMo benchmark will further improve this work. We will consider it in our future work. Thank you.
>
> ## Q7 Minor issues about the writing.
> Thank you for pointing these out. We have revised them in the revision.

---

> > ### Comment · Reviewer_55wR · 2022-08-07
> > **Reasonable responses**
> >
> > Thank you for the detailed response. I have read all the reviews and your response. I keep my evaluation. This work will be a good resource for the AutoML community, and also interesting to the MARL community.

---

### Official Review · Reviewer_fvNP · 2022-07-09

**Rating:** 5
**Confidence:** 4
**Soundness:** 2 fair
**Presentation:** 2 fair
**Contribution:** 2 fair

**Summary:**

This paper proposes to apply MARL to the dynamic algorithm conﬁguration (DAC) problem and instantiates it on a multi-objective optimization task. The method is mostly well presented and the experiments are illustrated clearly.

**Questions:**

1. Line 135-138: “higher rewards do not always imply better performance…”, I don’t understand what it means. How and why does your proposed reward function (equation 2) work?

2. Figure 1 and its corresponding texts are also very confusing. What do you mean by “repeated rewards”? Why switching from good to bad states can still get good rewards?

3. I cannot see any theoretical reason for using MARL on this problem. Specifically, I don’t know why DQN cannot work better than or at least as good as VDN?

4. How do you set the episode length?


**Limitations:**

1. The theoretical motivation of applying MARL on DAC problems is not convincing, even though the authors argue it by the heterogeneity of configuration hyperparameters. However, that is just a dimension problem, which doesn't mean single-agent RL cannot handle it.

**Strengths And Weaknesses:**

Strengths:
1. The researched question, i.e. applying MARL to DAC problems is interesting;
2. The experiments are well conducted;

Weaknesses:
1. The theoretical motivation to use MARL rather than single RL is not clearly analyzed;
2. The theoretical analysis for the different performance of the experiments is lacked;

---

> ### Author Response · Authors · 2022-08-02
> **Response to Reviewer fvNP (2/2)**
>
> ## Q3 Reward function
>
> Thank you for pointing out the issues of the presented reward function. Here we would like to take this opportunity to clarify why we ended up with such a reward function. In the following, we would like to first explain our motivation and then address your comments.
>
> Since the goal of an iterative optimization algorithm at iteration $t$ is to find the solution with the best objective function value so far rather than to just find a better solution than the solution at the $t-1$ step (since the quality of solutions in the last step may be poor), it would make sense to give the policy a reward when finding a better solution than the best solution so far (rather than finding a better solution than the solution at the last step).
>
> Another factor we may need to consider when designing a reward function is that with time, it is getting harder to find better-quality solutions. In the beginning, it may be easy to find rapid improvement of solution quality, but in later stages, improvement could be harder. As such, a reward function that rewards more an agent who can find a better solution in later stages can encourage the agent to find a very good solution in the end.
>
> Given the above, we adopted a triangle-based reward function (i.e., the area of the triangle). At step $t$, the current reward we obtain is the area of the triangle with side length $p_{t+1}$ minus the area of the triangle with side length $p_{t}$. The proposed triangle-based reward function has two properties: 1) it only rewards the agent when a new solution is better than the best solution found so far and 2) the reward increases in later stages.
>
> **Comment 1: what is ``repeated rewards" and why switching from good to bad states can still get good rewards.**
>
> Response 1: Apologies - we should have made this clearer. Here, we meant that the agent which generates a better solution at the $t+1$ step than the solution at the $t$ step will be given a reward, despite not being better than the best solution found so far. For example, let us say $f()$ from step $t$ to step $t+4$ are 1, 0.9, 1, 0.9, 1, respectively. The agent will be given rewards at the steps of $t+2$ and $t+4$ despite no better solution being found.
>
> **Comment 2: how and why does the reward function work.**
>
> Response 2: please see above the explanations of the motivation of presenting the reward function.
>
> **Comment 3: Figure 1 and the corresponding texts.**
>
> Response 3: In the revised paper, we have re-plotted Figure 1 and rewritten the Reward section to make them clearer.
>
> ## Q4 How to set the episode length?
>
> One generation in MOEA/D is one step in the contextual MMDP. Thus, we set the episode length as the maximum number of generations of MOEA/D, i.e., $m\times 100$, where $m$ is the number of the objectives. Note that $m\times 100$ is a common setting in multi-objective optimization.
>
> We hope that the above response can address your concerns. But if we missed anything, please let us know.

---

> ### Author Response · Authors · 2022-08-02
> **Response to Reviewer fvNP (1/2)**
>
>
> Thank you for your constructive comments and suggestions. Below please find our response.
>
> ## Q1 The motivation of using multi-agent RL rather than single-agent RL? Why DQN can not work better than or at least as good as VDN?
>
> Apologies - we should have made it clearer. For a large-scale task (e.g., many hyperparameters), the number of agents needed is large. When solving the task through a single-agent RL algorithm, with the growth of the number of agents, the action space of the agent exponentially increases, and thus it is hard for the single-agent RL algorithm (e.g., DQN) to find a good policy, particularly on those with heterogeneous properties. Using multi-agent RL algorithms effectively decomposes the action space, leading to a much lower action space for each agent. Consequently, it is much easier for each agent to find their own near-optimal policy [1-3].
>
> Moreover, for multi-agent RL it has been found (in the Individual-Global-Max (IGM) principle [4]) that the joint and individual greedy action selections have consistency, i.e., the individual best can result in the global best. Especially, the linear value factorization adopted by VDN (used in our work) can realize a sufficient condition for IGM principle [5-6], thus allowing for better performance than DQN on large scale control tasks.
>
>
> [1] Scaling multi-agent learning in complex environments. University of Massachusetts Amherst, 2011
>
> [2] A survey and critique of multiagent deep reinforcement learning. Autonomous Agents and Multi-Agent Systems, 2019.
>
> [3] Multi-agent reinforcement learning: A selective overview of theories and algorithms. Handbook of Reinforcement Learning and Control, 2021.
>
> [4] Qtran: Learning to factorize with transformation for cooperative multi-agent reinforcement learning. ICML, 2019.
>
> [5] Towards Understanding Cooperative Multi-Agent Q-Learning with Value Factorization. NeurIPS, 2021.
>
> [6] Understanding Value Decomposition Algorithms in Deep Cooperative Multi-Agent Reinforcement Learning. ArXiv, 2022.
>
>
> ## Q2 Analysis about DQN and MA-UCB? Why they are worse than MA-DAC?
>
> For the superior performance of MA-DAC over DQN, please see the response to Q1. MA-UCB uses multi-armed bandits (MAB) to online adjust the hyperparameters. MAB considers a much simpler setting than reinforcement learning (please see Chapter 2 in [1]), limiting its ability. MA-DAC, as a full reinforcement learning approach, can learn more information from training sets than MA-UCB and thus outperforms it.
>
> [1] Reinforcement learning: An introduction. The MIT press, 2018.

---

> > ### Comment · Reviewer_fvNP · 2022-08-09
> > **Response to the rebuttal**
> >
> > Thanks for the authors' detailed reply. They are valuable. However, I still have concerns, especially on the theoretical advantage of using MARL methods than other single-agent RL method on DAC problems:
> >
> > > When solving the task through a single-agent RL algorithm, with the growth of the number of agents, the action space of the agent >exponentially increases, and thus it is hard for the single-agent RL algorithm (e.g., DQN) to find a good policy, particularly on those with >heterogeneous properties. Using multi-agent RL algorithms effectively decomposes the action space, leading to a much lower action >space for each agent. Consequently, it is much easier for each agent to find their own near-optimal policy [1-3].
> >
> > I don't think the MARL algorithms like VDN or QMIX can avoid the increasing action spaces since they both have one centralized Q, which uses the joint actions also. The high dimension problem in single-agent RL also happens on MARL algorithms. The motivation of value decomposition methods is to enable decentralized execution, however, which in the DAC problems is not needed. The heterogeneity stuff should not be sufficient as the motivation to use MARL since single-agent RL can simply concatenate the action spaces, right?
> >
> > >Moreover, for multi-agent RL it has been found (in the Individual-Global-Max (IGM) principle [4]) that the joint and individual greedy >action selections have consistency, i.e., the individual best can result in the global best. Especially, the linear value factorization adopted >by VDN (used in our work) can realize a sufficient condition for IGM principle [5-6], thus allowing for better performance than DQN on >large scale control tasks.
> >
> > Based on my limited understanding, the IGM principle works by assuming the global Q is factorizable. However, how do you know the DAC problems are factorizable?
> >
> > Nonetheless, I am glad to see the empirical advantage of using MARL methods on multi-objective optimization problems. I would like to increase my score to boardline as I still can not see any theoretical motivation of MARL methods on this problem. If you can provide more convincing proof, I would like again to increase my score. Now while it seems other reviews are happy enough, I would still keep my concerns.

---

> ### Author Response · Authors · 2022-08-08
> **Dear Reviewer fvNP, are our responses address your questions?**
>
> Dear Reviewer fvNP:
>
> We thank you again for your comments and hope our responses could address your questions. As the response system will end in this Tuesday, please let us know if we missed anything. More questions on our paper are always welcomed. If there are no more questions, we will appreciate it if you can kindly raise the score.
>
> Sincerely yours,
>
> Authors of Paper10642

---

### Official Review · Reviewer_K9Qu · 2022-07-09

**Rating:** 6
**Confidence:** 4
**Soundness:** 2 fair
**Presentation:** 2 fair
**Contribution:** 3 good

**Summary:**

This work introduces the multi-agent dynamic algorithm configuration problem which models the hyperparameter search of an arbitrary iterative algorithm as a contextual multi-agent MDP. Each agent in the environment is tasked to determine the value of a specific hyperparameter with its actions corresponding to hyperparameter values, states indicating information about the tuned algorithm and rewards indicating performance. An instantiation of this problem is presented based on the MOEA/D evolutionary algorithm as the MaMo environment and an algorithm based on value decomposition networks (VDN) is trained and evaluated in this problem, and compared to using evolutionary and alternative (MA)RL algorithms.

**Questions:**

1. For experiments towards RQ2, it is unclear in which exact problems the policy was trained. L. 293 “... the policy is trained on the problems with 3, 5, and 7 objectives.” - does this include all problems with the respective number of objectives?
2. In line 302 it is stated that “Among all the policies, MA-DAC always takes the first or second place ...” - do you refer to MA-DAC (M) for this experiment?


**Limitations:**

The proposed problem of MA-DAC appears to have underlying assumptions which limit its applicability to new algorithms (outside of MOEA/D and similar iterative, evolutionary algorithms) and are insufficiently discussed.

**Strengths And Weaknesses:**

### Strengths
The primary contribution of this work is the formulation of the MA-DAC problem which models a hyperparameter search over multiple, heterogeneous hyperparameters as a multi-agent decision making problem. While the idea of dynamic algorithm configuration (DAC) is not novel, the framing as a multi-agent problem to the best of my knowledge is novel, interesting and significant.

### Weaknesses
However, the proposed framework lacks generality, the multi-agent algorithm presented alongside does not appear to contain novelty (outside of the modelled problem), and the evaluation is insufficient.

**Major:**
1. Lack of generality of MA-DAC problem modelling: The modelling of the hyperparameter search as a multi-agent decision making problem requires several properties from the underlying to-be-tuned algorithm which appear quite restrictive and are not sufficiently discussed.

    1.1. In particular, the tuned algorithm needs to be iterative with some metric that for each iteration gives a reasonable indication of its future performance. This is not true for e.g. most (MA)RL algorithms where future performance often only becomes visible after thousands if not millions of steps. One could model the entire training of the underlying algorithm as a single transition but in that case each transition would be too costly for this to be computationally feasible.

    1.2. Also, the modelling appears to assume that the tuned hyperparameters can be changed at each iteration of the algorithm. For some hyperparameters and algorithms, this is not possible (e.g. model size in DL).

2. The set of baselines in the experiments seems insufficient. The proposed approach based on the VDN MARL algorithm is only compared to a multi-agent version of UCB, a single-agent centralised controller trained with DQN and the evolutionary approach of MOEA/D. Given its similarity to other MARL algorithms, comparisons to independent learning algorithms such as IDQN, IPPO, and other CTDE learning algorithms like MAPPO, MADDPG and QMIX would be essential to understand the relevance and challenge of the proposed algorithm.

**Minor:**

3. The MA-DAC (M) algorithm is compared to in Table 4 but not included as a column - I strongly suggest to include this as a baseline in the table.
4. The authors suggest that “the importance of (tuned) hyperparameters varies” (l. 315f) because the setting of specific hyperparameters to “reasonable” default settings (l. 310) impacts performance differently. However, it is not clear whether these performances are obtained as a result of the relevance of the hyperparameters or the quality of the choice of the default setting for the different hyperparameters.

---

> ### Author Response · Authors · 2022-08-02
> **Response to Reviewer K9Qu**
>
> Thank you very much for carefully reviewing our paper and providing constructive comments and suggestion. Below please find our response.
>
> ## Q1 Lack of generality of MA-DAC problem modeling?
>
> We fully agree that in many RL tasks (e.g., playing football), the future performance only becomes visible after thousands of steps. It takes quite a while to know how good the policy is on such tasks. Having that said, there are also many RL tasks where the future performance can be obtained fairly quickly, such as in various optimization tasks where an indication of the future performance can be obtained immediately (i.e., just to evaluate newly-generated candidate solutions by the objective function(s)). On such tasks, DAC has been widely applied, e.g., adapting the learning rate in SGD, adapting the step-size in CMA-ES, and performing the heuristic selection in planning. Likewise, MA-DAC is also applicable to such tasks although it is designed to cope with more challenging ones (larger tasks, or with more heterogeneous properties).
>
> Regarding the changeability of hyperparameters, yes, you are right - some hyperparameters are not changeable at each iteration (e.g., model size in DL), which DAC (and MA-DAC) may not be able to be directly applied to.
>
> ## Q2 Insufficiency of only using VDN
>
> Thank you for pointing out this issue. Our main purpose of this work is to present an MA-DAC method to handle the dynamic configuration of algorithms with various types of hyperparameters. Thus VDN was used because of its simplicity and scalability. And we do find that under this algorithm, MA-DAC performs very well.
>
> Having that said, we fully agree that considering different algorithms is very helpful to understand the relevance and the challenge of the proposed work. Therefore, we have now added two MARL algorithms in the revised paper, an independent learning algorithm IQL and a CTDE learning algorithm QMIX (Appendix B.7), as suggested by the reviewer.
>
> We found (Table 12) that all of them perform better than DQN-1 (a more competitive baseline algorithm suggested by the Reviewer Y4Zz), but different MARL algorithms have the best performance on different instances (i.e., IQL on DTLZ2, QMIX on WFG4 and WFG7, and VDN on the rest). This occurrence may be due to the challenges caused by the randomness and heterogeneity of the tasks, which we believe will be beneficial in drawing the MARL community’s attention to these types of problems.
>
> ## Q3 The presentation of the table?
>
> We have added MA-DAC (M) as you suggested. We hope it is clear now.
>
> ## Q4 Default settings of the hyperparameters?
>
> There are four types of hyperparameters in our MOEA/D task: neighborhood size, weights, reproduction parameters, and reproduction operators. We configured their default setting by either following common practice/popular approach in the area of evolutionary multi-objective optimization or finding suitable setup empirically by ourselves, so that MOEA/D under the default settings already performs fairly well. Specifically, for the neighborhood size and the reproduction parameters, we used default settings in the area. For the weights, we used a popular, effective weight adaptation approach (i.e., MOEA/D-AWA). For reproduction operators, as different operators may be well-suited to different problems [1-2], we tested all the four types of operators (Appendix B.3) and chose the best-performing operator as our default one.
>
> [1] Adaptive operator selection with bandits for a multiobjective evolutionary algorithm based on decomposition. IEEE TEvC, 2013.
>
> [2] Deep reinforcement learning based parameter control in differential evolution. GECCO, 2019.
>
> ## Q5 Some minor comments
>
> **It is unclear in which exact problems the policy was trained. Does this include all problems with the respective number of objectives?**
>
> In all the experiments, the training sets are the three functions, i.e., DTLZ2, WFG4, and WFG6.
>
> **Do you refer to MA-DAC (M) for this experiment?**
>
> Yes. Thank you for pointing this out; we have revised it.
>
> We hope that the above response can address your concerns. But if we missed anything, please let us know.

---

> ### Author Response · Authors · 2022-08-08
> **Dear Reviewer K9Qu, are our responses address your questions?**
>
> Dear Reviewer K9Qu:
>
> We thank you again for your comments and hope our responses could address your questions. As the response system will end in this Tuesday, please let us know if we missed anything. More questions on our paper are always welcomed. If there are no more questions, we will appreciate it if you can kindly raise the score.
>
> Sincerely yours,
>
> Authors of Paper10642

---

> > ### Comment · Reviewer_K9Qu · 2022-08-08
> > **Response to Rebuttal**
> >
> > I thank the authors for their detailed rebuttals. I have read all reviews and rebuttals and have checked the updated paper and appendix. I particularly appreciate the authors efforts in comparing to different MARL algorithms (Appendix B.7), further experiments in Appendix C and many clarifications provided in the updated paper and rebuttals.
> >
> > With all the done changes, and after reading the other reviewers, I am largely in alignment with reviewers Y4Zz and 55wR's excitement for this work and increased my score.

---

### Official Review · Reviewer_Y4Zz · 2022-07-10

**Rating:** 8
**Confidence:** 5
**Soundness:** 3 good
**Presentation:** 3 good
**Contribution:** 3 good

**Summary:**

The work presents a multi-agent extension to the Dynamic Algorithm Configuration (DAC) framework. As such, the paper discusses how to extend the formulation of a contextual MDP to a contextual (cooperative) multi-agent MDP. Further, the work provides a benchmark in the form of MOEA/D which allows to study 1) dynamic algorithm configuration in the case of various heterogeneous hyperparameters, and 2) MARL algorithms.

**Questions:**

The questions are not ordered in any particular order.
* Why did you not consider any benchmarks from DACBench?
* As you target handling of different parameter types, how would you handle ordinal or categorical hyperparameters, as well as hyperparameters that are using some form of transformation (i.e. log-scaled learning-rates).
* Do you believe your approach is also helpful in the multi-hyperparameter case with hyperparameters of the same type?
* To me it seems that the proposed reward might give higher weight to crossing from positive function values to negative ones. Is my interpretation of that correct? If so, is there any particular benefit from that?
* How did you discretize the neighborhood size?
* How did you normalize the state features?
* How did you limit the frequency for adjusting the weights?
* Are all hyperparameters adjusted at the same frequency?
* How did you select the instances? What exactly are MOPs with similar properties?
* When testing, it is stated that "[...] all the functions are considered in the testing[...]" (line 266). Does that mean that you test on the training set?
* Why do you differentiate between three types of features in the state? Aren't indicators such as hypervolume and other statistics part of the optimization process?
* Why is the reward scaled by $\frac{1}{2}$?

**Limitations:**

The limitations are not particularly addressed. For example, a limitation of the work is that it only considers the multi-agent configuration on a single benchmark. Further, from the text it seems that the test set includes the training set which will result in inaccurate assessments on the generalization performance.

----

**During the rebuttal the authors remedied these shortcomings.**

**Strengths And Weaknesses:**

# Taking Author Feedback into account (during rebuttal)
I am very happy how the authors addressed my and other reviwers concerns. Thus **I already increased my score to strong accept**

-----
-----

# Original Review
Overall I'm very excited about this paper. I believe that dynamic configuration is an important problem that is not yet studied broadly enough. I am particularly excited about the extension to multiple hyperparameters of different types. In the following I will list the strengths and weaknesses.

-----
-----

## Strengths
The work combines two interesting and important fields of research (Multi-agent RL and DAC). This combination enables interesting research opportunities in both fields. For example, to the best of my knowledge, MARL methods have not had a particular focus on learning general policies. By following the DAC framework and using contextual MDPs (or rather contextual (cooperative) multi-agent MDPs) MARL algorithms can be evaluated with regards to their generalization capability. In particular, the notion of context allows to study generalization of policies in a principled manner (see e.g. [Kirk et al. 2021](https://arxiv.org/abs/2111.09794)), as was also discussed in the original work on DAC (referenced as [2] in the paper).
On the DAC side for example, as most methods employ some form of RL, the use of (cooperative) multi-agent methods allows to deal with very different types of hyperparameters (as shown in the paper). Thus I believe the combination of MARL and DAC is of importance to the NeurIPS community.

-----
-----

## Weaknesses
### Differentiation with DAC
The work calls the proposed multi-agent approach to DAC a novel framework, however I do not see large enough differences to talk about a novel framework. In particular, the original DAC work considers also tuning of multiple hyperparameters. As the go-to solution methods for DAC are vanilla deep RL approaches (e.g. DQN) the so far considered DAC scenarios are limited by the capabilities of vanilla RL, which typically does not deal with "heterogeneous" action spaces. Thus, from my point of view, the work proposes a novel *solution approach* to the DAC problem based on multi-agent RL, which might be superior as it allows to make efficient use of human knowledge (e.g. certain hyperparameters are distinct from each other).

To me, this weakness is particularly apparent in section 3.
For example in Section 3.1 (line 103-104) it is said that the work "... explicitly take[s] the instance distribution [...] as context into account to facilitate generalization." This same motivation was stated by Biedenkapp et al. 2020 (cited as [2] in the paper) when formulating the DAC problem as cMDP. The work presented here does not make this connection explicit.
Further, in Section 3.2 the work details the components of the cMMDP. In the paragraph about the state, desiderata are listed that are not unique to the multi-agent setting. For example, [Lindauer and Biedenkapp 2020](https://www.automl.org/wp-content/uploads/2020/08/AC_Tutorial_2020.pdf) presented a similar set of 3 desiderata for the state.

In Section 4.2 when the different hyperparameters are discussed, selection of different operators for differential evolution are mentioned. I believe in the main paper there should be a reference to Sharma et al. 2019 (cited as [10] in the appendix) as their work proposed to adapt DE operators via DQN.

Finally, although a benchmark collection for DAC problems, DACBench, is cited (as [9] in the paper), none of the existing benchmarks were used for evaluation. This seems strange to me as I believe there are benchmarks in the collection that allow for configuration of multiple hyperparameters as well as multiple hyperparameters of different types. At least the former seems to be covered as the artificial benchmarks from the ECAI paper by Biedenkapp et al. 2020 are contained. It would have given a good experiment that could have shown if a multi-agent approach can outperform  the DQN results by Biedenkapp et al.

Thus, overall I believe the work would benefit if from a closer alignment with DAC.

-----

### Notation
The work is somewhat inconsistent with the notation. In Section 2.1 contextual MDPs are discussed, though context is not formally introduced. Later in Section 3.1 context is denoted as $i\in\mathcal{I}$, however the connection to problem instances is not given. Later on when the transition function is discussed the connection between context $i$ and problem instances $f\in\mathcal{F}$ is not obvious. Thus for readers not familiar with DAC it might make it difficult to follow what context is and how it is used in this work. Further, the index variable $i$ is overloaded. On line 82 it is first introduced as the agent index and later reused to denote context. Thus in line 126 it was at first confusing as it seemed that the actions were context dependent rather than $a^{(i)}$ giving the agent index.

The way the reward function in equation 2 is written up is rather confusing. I would suggest to expand the $p^2_t$ variables and then to rearrange similar to
$$r_t = -0.5\cdot \frac{
 (f_{t}^{\star} - f_{t-1}^{\star}) \cdot ( f_{0}^{\star} - f_{t}^{\star} + f_{0}^{\star} -  f_{t-1}^{\star})
}{
(f_0^{\star})^{2}
}$$
where $f_{t}^{\star}$ denotes the best observed function value up til time-step $t$ (thus, $f_{0}^{\star} = f(s_0)$).
I think this highlights better that the reward function only reflects changes in the incumbent values. Also, to me, it better highlights how improvements over the default function value are rewarded (and how the reward is scaled).
In regards to that, I do not understand how Figure 1 should be interpreted. Since the reward is solely based on changes in incumbent, the reward is more sparse than Fiugre 1 suggests. While in the beginning it is reasonable to expect rapid improvements in incumbents, I believe in later stages improvements will be more infrequent, thus often leading to rewards of 0. Thus, overall I would expect more plateaus in Figure 1.

-----

### Claim about instance sets
It is stated that "[...] instances should have similar properties, otherwise the good generalization ability is hardly achievable [9]" (lines 156&157). That claim is false and the citation (to DACBench) does not seem correct. Biedenkapp et al. 2020 (cited as [2] in the paper) proposed artificial benchmarks on which one can control the heterogeneity of the problem instances. They showed that by using context information it is possible to learn policies even across very heterogeneous problem instances. Further, Shala et al. 2020 (cited as [29] in the paper) showed that DAC policies for step-size adaptation of CMA-ES are capable of generalization to even dissimilar functions. This is possible as context information enables RL agents to learn to distinguish between problem instances and learn about how they relate to each other.
Thus I believe the claim is wrong.

-----

### Experiments
I believe the experiments would be strengthened by showing results on the artificial benchmarks of Biedenkapp et al. 2020 as they used a DQN to control multiple hyperapameters (of the same type) the same way the DQN baseline in this work was used. This could highlight the importance of a multi-agent approach over a single-agent approach already for larger (but homogeneous) action spaces. I believe this would be a particularly informative experiment as the benchmarks allow to arbitrarily scale the number of parameters and number of choices per parameter. This could neatly highlight the advantage of a multi-agent approach.

On the evaluated benchmarks, the DQN baseline actually does not make too much sense to me. On the real benchmark, the action space is 128. Biedenkapp et al. showed that DQN does not scale well with the number of parameters, even on the artificial benchmarks. In a more recent work by [Biedenkapp et al. 2022](https://arxiv.org/abs/2202.03259) showed that on more stochastic DAC problems, DQN already struggled to scale with the number of hyperparameter value choices of a single hyperparameter. In their experiments, the DQN already struggled to perform well with only 10 values for the single hyperparameter.
I would suggest to change this baseline to a DQN that only changes the most important parameter (according to the ablation analysis from the paper). This would give the DQN at most an action space of size 4, which is much easier to learn. I think that would better highlight the importance of tuning all hyperparameters and provide a likely more competitive baseline.

Some of the experimental setup is not immediately clear without the appendix. It would be beneficial for the reader to move parts of the appendix, that describe the experimental setup in more detail, to the main paper. To make room for this I would suggest to move some of the details of Tables 3 or 4 to the appendix instead and only report average ranks and the "$+$/$-$/$\approx$" row.

-----
-----

## Overall
I am very enthusiastic about the presented work. I believe it provides an interesting extension to the DAC framework which will be interesting to researchers of AutoML/DAC but also for researchers of multi-agent RL. I believe the weaknesses I listed above should be addressed. (I think they can be addressed without too much effort). For now I vote for weak acceptance but I am willing to change my score based on the discussion with the authors and other reviewers.

-----
-----


## References
* [Kirk et al. 2021](https://arxiv.org/abs/2111.09794);  "A Survey of Generalisation in Deep Reinforcement Learning"; https://arxiv.org/abs/2111.09794
* [Lindauer and Biedenkapp 2020](https://www.automl.org/wp-content/uploads/2020/08/AC_Tutorial_2020.pdf); "Algorithm Configuration
Challenges, Methods and Perspectives"; Tutorial at PPSN 2020 and IJCAI 2020, https://www.automl.org/wp-content/uploads/2020/08/AC_Tutorial_2020.pdf
* [Biedenkapp et al. 2022](https://arxiv.org/abs/2202.03259); Theory-inspired Parameter Control Benchmarks for Dynamic Algorithm Configuration; https://arxiv.org/abs/2202.03259

---

> ### Author Response · Authors · 2022-08-02
> **Response to Reviewer Y4Zz (4/4)**
>
> ## Q11 How did you select the instances? What exactly are MOPs with similar properties?
>
> There are many types of properties in multi-objective optimization problems, such as separability, modality, bias, and geometry (Table 5 in [1]). We select the instances with concave geometry, among which we randomly select DTLZ2, WFG4, and WFG6 as the training sets.
>
> [1] A large-scale experimental evaluation of high-performing multi- and many-objective evolutionary algorithms. Evolutionary computation, 2017.
>
> ## Q12 Why do you differentiate between three types of features in the state?
>
> The first part of state features is used to describe the general properties of the optimization problems.
>
> The second part of state features is to emphasize the general information of the algorithm, i.e., the computational budget that has been used and the stagnant count ratio.
>
> To show the properties of the population and that how the population is evolved, we use some indicators and their statistic metrics in the third part of state features. We are sorry that we did not make the motivation of the three types of features clear in the original paper.
>
> We have revised it in the main paper and Appendix A.2.
>
> ## Limitation: Inaccurate assessments on the generalization performance.
>
> We trained on DTLZ2, WFG4, and WFG6, and tested on the other functions. In the tables of the original paper, the comparison is based on all of them. Thanks to your suggestion, we have revised to compare the methods on the training and testing functions, respectively. The results are shown in the last two rows of the tables in the revised version. It can be observed the proposed MA-DAC has clear advantage over the other methods on both the training and testing functions. Thank you.

---

> > ### Comment · Reviewer_Y4Zz · 2022-08-03
> > **I am increasing my score**
> >
> > Thank you very much for addressing my feedback so detailed. I have also read all other reviews and your responses to these and I have checked the changes that you highlighted in the paper as red text. I am happy with these changes and believe the paper has been much improved. I have already incresed my score to strong accept.

---

> ### Author Response · Authors · 2022-08-02
> **Response to Reviewer Y4Zz (3/4)**
>
> ## Q7 Presentation of the experiments
>
> Thanks for your suggestion. But we find that even moving some parts of the experiments to the appendix, the space is still not sufficient for including the detailed experimental setup. Due to time limit, we have not found a good way to improve this point. We will try our best to add more details about the experimental setup in the main body of the final version.
>
> ## Q8 How to handle ordinal, categorical, or transformed hyperparameters?
>
> For any type of hyperparameters, we only need to design proper action spaces. For example, for a hyperparameter with $d$ categories, it can be formulated as an agent with $d$-dimensional discrete action space.
>
> ## Q9 Is MA-DAC helpful in the multi-hyperparameter case with hyperparameters of the same type?
>
> The added experiment on the Sigmoid problem in DACBench according to your suggestions has shown that MA-DAC can be helpful.
>
> ## Q10 Details about the experiments
>
> **Comment 1: how did you discretize the neighborhood size?**
>
> Response 1: The default neighborhood size is $20$, which is a common setting. We use the other three values close to $20$, i.e., $15$, $25$ and $30$, to discretize the neighborhood size. Thus, the neighborhood size is selected from $\{15, 20, 25, 30\}$.
>
> **Comment 2: how did you normalize the state features?**
>
> After checking the state features, we find that the statement ``Note that all the states are normalized to $[0,1]$ to ensure the generalization of the learned MA-DAC policy." in the original paper is not exact. We are sorry for our carelessness. Although we pre-processed some state features to facilitate the generalization, they do not fall strictly in $[0,1]$. We have modified the statement in Section 4.2, and added a detailed description about how we pre-processed the state features in Appendix A.2.
>
> For state features 0 and 1, we use $1/m$ and $1/D$, respectively, where $m$ is the number of objectives and $D$ is the number of variables. They obviously belong to $[0,1]$. For state features 2, 3, 4, and 5, they are defined in $[0,1]$. State feature 6 is the average distance between the solutions in the population. We sample a sufficient number of solutions before optimization, and calculate the maximum distance between them, which is used as a scaling denominator when calculating state feature 6. State features 7-9 are the difference of two values in $[0,1]$, and thus belong to $[-1,1]$. For state features 10-21, they are statistical values (i.e., mean and standard deviation) of state features 4-6.
>
> **Comment 3: how did you limit the frequency for adjusting the weights?**
>
> Response 3: We use 5\% of total steps in one episode based on the setting in [1].
>
> [1] What Weights Work for You? Adapting Weights for Any Pareto Front Shape in Decomposition-Based Evolutionary Multiobjective Optimisation. Evolutionary Computation, 2018.
>
> **Comment 4: are all hyperparameters adjusted at the same frequency?**
>
> Response 4: All the hyperparameters except the adaptive weights can be adjusted at each step.

---

> ### Author Response · Authors · 2022-08-02
> **Response to Reviewer Y4Zz (2/4)**
>
> ## Q4 Reward function
>
> Thank you for pointing out the issues of the presented reward function. Here we would like to take this opportunity to clarify why we ended up with such a reward function. In the following, we would like to first explain our motivation and then address your comments.
>
> Since the goal of an iterative optimization algorithm at iteration $t$ is to find the solution with the best objective function value so far rather than to just find a better solution than the solution at the $t-1$ step (since the quality of solutions in the last step may be poor), it would make sense to give the policy a reward when finding a better solution than the best solution so far (rather than finding a better solution than the solution at the last step).
>
> Another factor we may need to consider when designing a reward function is that with time, it is getting harder to find better-quality solutions. In the beginning, it may be easy to find rapid improvement of solution quality, but in later stages, improvement could be harder. As such, a reward function that rewards more an agent who can find a better solution in later stages can encourage the agent to find a very good solution in the end.
>
> Given the above, we adopted a triangle-based reward function (i.e., the area of the triangle). At step $t$, the current reward we obtain is the area of the triangle with side length $p_{t+1}$ minus the area of the triangle with side length $p_{t}$. The proposed triangle-based reward function has two properties: 1) it only rewards the agent when a new solution is better than the best solution found so far and 2) the reward increases in later stages. The suggested reward function is essentially equivalent to our reward function, but we keep the triangle-based representation for better illustrating our motivation.
>
> Comment 1: why is the reward scaled by 1/2.
>
> Response 1: As the reward is defined by the area of the triangle, 1/2 is just to ensure the geometry meaning.
>
> Comment 2: the reward may be sparse.
>
> Response 2: The experimental results show that our proposed reward is not sparse. We plotted the IGD curves in Appendix B.5 (i.e., Figures 2, 3, and 4) in the original paper. As shown in the figures, the IGD value is continuously declining even in the later stages, implying that the reward is not 0. We agree that the reward may be sparse when there are more iterations on MaMo, or on other types of problems. We will try to address this issue in our future work by using some techniques (e.g., intrinsic reward). Thank you for your suggestions.
>
> Comment 3: higher weight to crossing from positive to negative.
>
> Response 3: Since the reward is the area of the triangle, a higher reward is given to a solution which is farther away from the initial solution (i.e., $f(s_{0})$). For example, let us say $f(s_{0}) = 10$, $f_t^* = 6$, $f(s_{t+1}) = 2$, and $f(s_{t+2})=-2$. Suppose that the smaller the value of $f$, the better the solution. The reward $r_{t+1}$ is $6/25$, while the reward $r_{t+2}$ is $10/25$. In our experiments, the function (i.e., IGD) value has a minimum of $0$, and will not cross from positive to negative values.
>
> We hope that the above explanation has made the reward function clear. But if we have missed some concerns, please let us know.
>
> ## Q5 Claim about the instance sets
>
> Good point. Yes, context information can enable RL agents to learn to distinguish between problem instances and learn about how they relate to each other, thus allowing for more generalized policies that are able to deal with problem instances with even very heterogeneous properties.
>
> Apologies - we should have been more clearer in the original paper that we did not mean that DAC cannot generalize to heterogeneous properties but that DAC may generalize better with similar properties. This has also been found in our experiments, e.g., the same number of objectives (Table 3).
>
> In this revision, we have rectified the statement to make this issue clearer.
>
> ## Q6 Experiments on DQN that only changes the most important hyperparameter
>
> Thanks to your suggestion, we have revised to add the comparison with DQN that dynamically adjusts only one type of hyperparameters of MOEA/D. There are four DQN variants, denoted as DQN-1, DQN-2, DQN-3, and DQN-4, representing using DQN to adjust weights, neighborhood size, types of reproduction operators, and parameters of the reproduction operators, respectively. The results are shown in Appendix B.6. As expected, we can observe from Table 11 that DQN-1 performs the best among the four DQN variants, because adaptive weights are generally more important (as shown in Table 4). Note that to address the Q2 by Reviewer K9Qu, we have implemented MA-DAC with different MARL algorithms. Table 12 shows that all the three MA-DAC policies outperform DQN-1, demonstrating the effectiveness of MA-DAC.

---

> ### Author Response · Authors · 2022-08-02
> **Response to Reviewer Y4Zz (1/4)**
>
> Thank you very much for carefully reviewing our paper and providing constructive comments and suggestions, which have helped improve the work a lot. We are very glad that you appreciate our work. We hope that our response has addressed your concerns, but if we missed anything please let us know.
>
> ## Q1 Differentiation with DAC
>
> We are sorry that we did not clearly discuss the relationship between MA-DAC and DAC. We fully agree that MA-DAC proposes a multi-agent extension to the DAC framework, and should be seen as a novel *solution approach* to DAC based on multi-agent RL, rather than a novel *framework*. We have revised the corresponding statement in the paper. We have also revised to make the connection to previous works more clearly and add the missed references according to your suggestions. Thank you very much.
>
> ## Q2 Experiments on DACBench
>
> We used the dynamic configuration of MOEA/D as our benchmark, because this complex and well-known multi-objective optimization algorithm has multiple heterogeneous types of hyperparameters. But your suggestion is a very good idea. The empirical comparison on DACBench can be used to evaluate the performance of MA-DAC more comprehensively.
>
> We have revised to add the experiment on the Sigmoid problem in DACBench, which allows the configuration of any number of homogeneous hyperparameters. In particular, we use $5D$-Sigmoid and $10D$-Sigmoid(i.e., there are $5$ and $10$ agents in MA-DAC, respectively) with action space size $3$ (i.e., each agent has $3$-dimensional discrete action space). The baselines we use are static policy and DQN policy. For MA-DAC, besides VDN (i.e., the default MARL algorithm employed in this work), we also use IQL and QMIX to examine the performance of MA-DAC with different MARL algorithms, as suggested by Reviewer K9Qu. The results are shown in Figure 5 in the revised paper. Here we give a simplified table for clear presentation. Each result consists of the mean and standard deviation of 10 runs.
>
> |             | Static        | DQN           | IQL           | VDN           | QMIX          |
> | ----------- | ------------- | ------------- | ------------- | ------------- | ------------- |
> | 5D-Sigmoid  | 1.360 (0.186) | 1.953 (0.232) | 5.372 (0.173) | 6.098 (0.212) | 6.178 (0.211) |
> | 10D-Sigmoid | 0.056 (0.003) | —             | 0.708 (0.347) | 1.326 (0.329) | 1.981 (0.625) |
>
>
> It can be observed that DQN cannot achieve a high reward on $5D$-Sigmoid, which is consistent with the previous observation [1]; while all the three MA-DAC versions can achieve significantly better performance than DQN and static policy. Even on $10D$-Sigmoid (Figure 5(b)), the MA-DAC policies still perform well, showing the scalability of MA-DAC. Thus, MA-DAC can not only handle large-scale heterogeneous tasks (e.g., the MaMo benchmark used in the orginal paper), but also large-scale homogeneous tasks (e.g., the $10D$-Sigmoid benchmark added according to your suggestion). Thank you.
>
> [1] Dynamic algorithm configuration: Foundation of a new meta-algorithmic framework. ECAI, 2020.
>
> ## Q3 Notations are inconsistent
>
> We have revised the notations about the contexts, instances and actions, to make them clear. Thanks for your comments.

---

### Author Response · Authors · 2022-08-02
**General response to reviewers**


We appreciate valuable comments from all reviewers. We have revised the paper carefully according to your suggestions. For the sake of clarity, the revisions have been colored in red. The modification parts are briefly summarized as follows.

- Introduction.
    - We have revised the presentation of MA-DAC, to make it more accurate.
- Experiments.
    - We have added more experiments, including comparison with DQN that tunes only a single type of hyper-parameters, comparison among different MARL algorithms, and comparison on DACBench.
    - We have revised the presentation of the experiments, to make them more clear.

- Writing.
  - We have revised the typos and other issues mentioned by all the reviewers.

We hope that our response has addressed all the questions and concerns. But if we missed anything, please let us know. We are always willing to answer any of your concerns about our work and we are looking forward to the following inspiring discussions.

---

> ### Author Response · Authors · 2022-08-03
> **A new version of the paper is uploaded**
>
> Dear reviewers, we have uploaded a new version of the paper and appendix. Please see this new one. We are very sorry for any inconvenience this may have caused.

---

> ### Author Response · Authors · 2022-08-09
> **Results update**
>
> Dear Reviewers:
>
> Just a quick update that the experiments on the functions with the other numbers of objectives have just been finished. This is for Appendix B.6 (comparison with different DQN variants) and Appendix B.7 (comparison with different MARL algorithms), suggested by Reviewers Y4Zz and K9Qu.
>
> The results remain the same. That is, for the comparison with different DQN variants, the DQN variants that dynamically adjust one type of hyperparameter perform better than the DQN that dynamically adjusts all the four hyperparameters at the same time. For comparison with different MARL algorithms, the three MA-DAC policies perform better than the baseline DQN, and among these three MA-DAC policies, no one performs best on all the instances.
>
> We have updated the paper accordingly.
>
> Finally, we would like to thank you again for providing constructive comments and suggestions, which have helped improve the paper a lot.

---

### Meta-Review · Area_Chair_fTcx · 2022-08-25

**Recommendation:** Accept
**Confidence:** Certain

**Metareview:**

After reading the reviews, feedbacks and discussions, I lean towards acceptance. Reviewers are all in favour of acceptance with different levels of enthusiasm. Some reviewers found particularly interesting that MARL methods could be applied to DAC problems but other reviewers were not entirely convinced why MARL is more suited that single RL or even classical optimization techniques for this particular problem. However, this could generate an interesting discussion between the DAC and MARL communities and this is the main reason why I vote for acceptance. The authors should consider adding a discussion section explaining why MARL could be fundamentally more advantageous/general than single RL methods in addition to their exhaustive experimental section.

**Award:**

No

---

### Decision · Program_Chairs · 2022-09-14

Accept